# Estrogen Receptors Alpha and Beta Mediate Synaptic Transmission in the PFC and Hippocampus of Mice

**DOI:** 10.3390/ijms22031485

**Published:** 2021-02-02

**Authors:** Mingyue Zhang, Hannah Weiland, Michael Schöfbänker, Weiqi Zhang

**Affiliations:** Lab for Molecular Neuroscience, Clinic for Mental Health, University Hospital Muenster, 48149 Muenster, Germany; weil02@uni-muenster.de (H.W.); m_scho79@uni-muenster.de (M.S.); wzhang@uni-muenster.de (W.Z.)

**Keywords:** 17ß-Estrdiol, ERα, ERß, sex difference, synapse

## Abstract

Distinct from ovarian estradiol, the steroid hormone 17ß-estradiol (E2) is produced in the brain and is involved in numerous functions, particularly acting as a neurosteroid. However, the physiological role of E2 and the mechanism of its effects are not well known. In hippocampal slices, 17ß-estradiol has been found to cause a modest increase in fast glutamatergic transmission; because some of these effects are rapid and acute, they might be mediated by membrane-associated receptors via nongenomic action. Moreover, activation of membrane estrogen receptors can rapidly modulate neuron function in a sex-specific manner. To further investigate the neurological role of E2, we examined the effect of E2, as an estrogen receptor (ER) agonist, on synaptic transmission in slices of the prefrontal cortex (PFC) and hippocampus in both male and female mice. Whole-cell recordings of spontaneous excitatory postsynaptic currents (sEPSC) in the PFC showed that E2 acts as a neuromodulator in glutamatergic transmission in the PFC in both sexes, but often in a cell-specific manner. The sEPSC amplitude and/or frequency responded to E2 in three ways, namely by significantly increasing, decreasing or having no response. Additional experiments using an agonist selective for ERß, diarylpropionitrile (DPN) showed that in males the sEPSC and spontaneous inhibitory postsynaptic currents sIPSC responses were similar to their E2 responses, but in females the estrogen receptor ß (ERß) agonist DPN did not influence excitatory transmission in the PFC. In contrast, in the hippocampus of both sexes E2 potentiated the gluatmatergic synaptic transmission in a subset of hippocampal cells. These data indicate that activation of E2 targeting probably a estrogen subtypes or different downstream signaling affect synaptic transmission in the brain PFC and hippocampus between males versus females mice.

## 1. Introduction

The classic mechanisms of estrogen, usually in the form of estradiol, involve activating the nuclear receptors estrogen receptor alpha (ERα) and beta (ERß), which subsequently affect the regulation of gene expression, neuroprotection, and neural growth [1,2]. However, early experiments have shown that estrogens can acutely affect synaptic responses via nongenomic actions, and these rapid effects are thought to result from the activity of membrane-associated estrogen receptors upon activation by selective receptor agonists [3,4,5]. Recently, the neurosteroid 17ß-estradiol (ERß) was found to be produced at high levels within the brain of both males and females. Additionally, previous studies have shown that classic estrogen receptor proteins are expressed not only in the nucleus but also at extranuclear sites, including at synapses [6].

Nonetheless, it is still unclear which ER subtypes are localized to which areas of the brain. Previous studies on the location of ERα and ERß seem to have contradicting results, possibly because the studies used different animal species, different antibodies or even different immunohistochemical methods [7,8,9]. Further, only limited data is available for the area of the prefrontal cortex (PFC)—an area of the brain that is tightly connected to limbic formation and is dedicated to memory, planning and executing actions [10,11]—but initial studies have suggested that the PFC contains nERα/ß and mERα/ß [12]. Thus, it may respond to ER agonists.

For decades researchers have known that estrogen acutely alters the intrinsic excitability of neurons in the hypothalamus/preoptic area, amygdala, striatum, cerebellum and hippocampus in both sexes [13,14,15]. In most of these experiments, estradiol was found to alter neuronal firing rates and/or modulate the resting membrane potential or limit action potentials in vivo or in vitro. In addition, estrogens can reduce Ca^2+^ currents in brain areas within minutes [16]. Although the mechanisms of these actions are unclear, more and more evidence is accumulating. Recently, the 17ß-estradiol (E2)-synthesizing enzyme P450 aromatase was found to be expressed in hippocampal neurons and to synthesize estrogen as a neurosteroid; these studies have suggested that E2 can acutely modulate synaptic function in vivo [17,18]. Locally synthesized estradiol was also found to affect nearby neurons and synapses in a paracrine or autocrine way through neuronal activity (Hojo Y et al., 2004). It remains to be seen whether E2 affects other brain areas, such as the PFC, and whether such effects are sex specific.

Any sex differences in the nervous system may be latent differences, in which identical functions in males and females are achieved through different underlying mechanisms [19]. This is evidenced by the fact that many brain disorders vary in males and females. However, most such previous studies involved only males, as it was assumed that female brains are more variable [20]. In this study, we examined both male and female mice and report sex differences in the regulation of excitatory synapses, GABAergic inhibition in the hippocampus, and activation of ERß-mediated synaptic activity in the PFC.

## 2. Materials and Methods

The experiments were performed in accordance with the European Communities Council Directive (86/EEC) and were approved by the Federal State Office for Consumer Protection and Food Safety of North Rhine-Westphalia, Germany. Every effort was made to reduce the number of animals used in the experiments. All mice were given ad libitum access to water and food and were housed under a 12 h light/dark cycle. For analyzing ex vivo isolated but functional neuronal networks, brain preparations including the mPFC from 8- to 12-week old C57BL/6 mice were employed.

### 2.1. Slice Preparation and Whole-Cell Recordings

The slice preparation was performed as described previously [21]. Briefly, after quick decapitation, mice brains were transferred to ice-cold oxygenated artificial cerebrospinal fluid (ACSF). Then, 300-µM-thick slices containing the PrL and IL or hippocampus were cut on a vibratome (VT 1200, Leica, Germany) and obtained as previously described. Slices were placed in the recording chamber, which was perfused (4 mL/min) with ACSF at room temperature.

In slices of hippocampus, all recordings (described in detail below) were performed on pyramidal neurons in CA1, as described previously [22]. In slices of PFC, all recordings were conducted in layer 2/3 of the prelimbic cortex and infralimbic cortex. The pyramidal neurons were identified by their morphologies under infrared differential interference contrast (IR-DIC) visualization. The glass microelectrodes varied between 3–5 MΩ. Access resistance was monitored before recording and after recording using transient current responses to hyperpolarization 5 mV pulses and exhibiting a change in access resistance >10% were excluded from the analyses. We excluded patches with a serial resistance of >20 MΩ, a membrane resistance of <0.8 GΩ, or leak currents of <150 pA. The membrane currents were filtered by a four-pole Bessel filter at a corner frequency of 2 KHz and digitized at a sampling rate of 5 KHz using the DigiData 1322 A interface (Axon Intruments/Moleclar Devices, Sunnyvalem, CA, USA).

The spontaneous excitatory postsynaptic currents (sEPSCs) were measured in acute coronal PFC slices from prelimbic cortex or from hippocampus CA1 neurons at a holding potential of −70 mV. sEPSC recordings were taken from pyramidal neurons in PrL layer 2/3 in the presence of strychnine (a glycine receptor antagonist; 5 µM) and 1(S), 9(R)(−)-bicuculline methochloride (a competitive GABA_A_ receptor antagonist; 5 µM unless otherwise indicated). The spontaneous inhibitory postsynaptic currents (sIPSCs) were recorded in the presence of 6-cyano-7-nitroquinoxaline-2,3-dione (CNQX; 10 µM) and DL-2-amino-5-phosphonopentanoic acid (DL-AP5; 50 µM). For sIPSC measurements, the recording pipettes were filled with a solution containing (in mM): 140 KCl, 1 CaCl_2_, 10 EGTA, 2 MgCl_2_, 0.5 Na_2_-GTP, 4 Na_2_-ATP, and 10 HEPES (pH was adjusted to 7.2 with KOH). To record sEPSCs, the pipettes (input resistance: 3–5 MΩ) were filled with the following solution (in mM): 140 potassium gluconate, 1 CaCl_2,_ 10 EGTA, 2 MgCl_2_, 4Na_3_ATP, 0.5 Na_3_GTP, 10 HEPES, pH 7.3. The bath solution in all experiments consisted of 125 NaCl, 2.5 KCl 1.25 Na_2_HPO_4_, 2 MgSO_4_, 26NaHCO_3_, 1.5 CaCl_2_, 14 glucose (pH 7.4, aerated with 95% O_2_, 5% CO_2_). To examine the activation of estrogen receptor mediated neurotransmission, E2 (ß-Estradiol, 100 nM, Tocris Bioscience, Bistrol, UK) and DPN (Tocris Bioscienc, Bistrol, UK, 10 nM) were bath applied. All other chemicals were obtained from Sigma-Aldrich (St. Louis, MO, USA).

### 2.2. Kinetics Analysis

For decay kinetic study, the decay time (τ) constant was determined by fitting an exponential function to the falling phase of the group of average mPSC from 90% to 10% of its peak amplitude [23]. The decay of mPSCs of sample average was adequately fitted by a two-exponential curve. The two exponential decay function are described by the following equation:*I*(*t*) = *I*_f_ exp(−*t*/τ*_f_*) + *I*_s_ exp(−*t*/τ*_s_*)(1)

*I* = amplitude; τ = decay time constant; *f* = fast, *s* = slow component.

The decay time of averaged current was conducted with a least-squares method using exponential fitting routine. In most cases, for comparing decay time between different experimental conditions, a combination between the two decay time components into the weighted average determined using the following equation [24]:τw = [I_f/_(I_f_ + I_s_)]×τ_f_ + [Is/(I_f_ + I_s_)] × τ_s_(2)

### 2.3. Data Analysis

The software programs Mini Analysis 6.0.3 (Synaptosoft, Decatur, GA) ClampFit 10.3 (Axon Intruments/molecular Devices), Prism 5 (GraphPad software, San Diego, CA, USA), and IBM SPSS Statistics were used for data evaluation and further statistical processing. The detection threshold of spontaneous excitatory events was set at twice the baseline noise. To exclude false events, each measurement was visually inspected and analyzed. Data are presented as mean ± standard error of the mean (SEM). Shapiro–Wilk tests were used to determine data for normally and non-normally distributed data in the individual cell analysis. Parametric unpaired Student’s *t*-test or nonparametric Mann–Whitney test were used to determine differences between baseline and drug-treated samples for normally- and non-normally distributed data, respectively. The level of statistical significance was set α = 0.05. Statistical significance is indicated as an * for *p* < 0.05.

## 3. Results

### 3.1. Estradiol Affects sEPSC Amplitude and Frequency in a Subpopulation of PFC Cells in Female Mice

To study the effect of E2 on excitatory transmission, we first measured sEPSCs in the presence of E2 (E2:100 nM) in the PFC of female mice. All recordings were measured in pyramidal neurons in the PFC by whole-cell recording. These recordings showed that the sEPSC amplitude responded to E2 in different cells. The overall effect of E2 was to increase the sEPSC in the PFC by 4%, but these results was with no variation (Figure 1a Female; Base: 14.43 ± 0.56 pA vs. 14.87 ± 0.55 pA; U = 84.5, Z = −0.61 *p* = 0.54, *n* = 14). However, within-cell analyses showed that 3 of 14 cells experienced significant sEPSC amplitude increases, ranging from 8% to 18%, in response to E2, but the remaining cells showed no effect in response to E2 (Figure 1b). Thus, the proportion of cells that increased their sEPSC amplitude in response to E2 was 21% (Figure 1e). Similar to the PFC response to E2 in sEPSC amplitude, overall E2 did not cause the PFC cells to show changes in sEPSC frequency (Figure 1c Base: 2.38 ± 0,45; E2: 2.28 ± 0.41; U = 103, Z = 0.21 *p* = 0.836). The within-cells analyses showed that variations only 1 of the 14 cells examined had a statistically significant drop in sEPSC frequency in response to E2, and another 1 cell had a significant increase in sEPSC frequency (Figure 1d). The proportion of cells that showed a significant increase in sEPSC frequency in response to E2 was 7%, and the proportion that showed a significant decrease in sEPSC frequency was also 7% (Figure 1f). As the sample size was small, we cannot determine whether it was indeed the effects on the cells, or some other mechanisms and this change needs further study. Moreover, plotting the normalized change in sEPSC frequency versus amplitude showed that significant changes in frequency and amplitude were most likely to occur in different cells (Figure 1h). Overall, we found no correlation between the relative frequency and relative amplitude changes of sEPSC in response to E2 (Figure 1h).

### 3.2. Estradiol Leads to Opposing Reactions in sEPSC Frequency in a Subset of PFC Cells in Male Mice

We next addressed whether the excitatory synaptic transmission is activated by E2 in males and whether this differs between females and males. For male mice, the overall effect of E2 on sEPSC amplitude was small, as it only increased by 6% (Figure 2a Base: 15.07 ± 0.43 pA, E2: 15.96 ± 0.70 pA, *n* = 24) and these results were with no variation (Figure 2a U = 75, Z = −1.04, *p* = 0.30 *n* = 24). Within-cell analyses showed that 12 of 24 cells had increased sEPSCs amplitudes ranging from 4% to 47%, an increase that was significant in 10 of 24 cells; conversely, only 1 cell showed a significantly decreased sEPSC amplitude compared to baseline (Figure 2b). As with amplitude, the overall effects of E2 on sEPSC frequency were small, with an overall increase of only 2%. (Figure 2c Base: 3.626 ± 0.35 Hz, E2: 3.685 ± 0.44 Hz, U = 300, Z = 0.24, *p* = 0.81 *n* = 24). Within-cell analyses showed that E2 significantly increased the sEPSc frequency in 7 of 24 cells, in which increases ranged from 13% to 145%, the whereas the sEPSc frequency significantly decreased in 8 of 24 cells, with decreases ranging from 17% to 50% (Figure 2d). Thus, the proportion of male PFC cells in which sEPSC amplitude increased in response to E2 was 42% and decreased in response to E2 was 4% (Figure 2e); the proportion of cells in which sEPSC frequency increased in response to E2 was 29% and decreased in response to E2 was 33% (Figure 2f). Again, no clear correlation was found between relative frequency and relative amplitude changes (Figure 2h).

### 3.3. Activation of ERß Has No Influence on Excitatory Transmission in the PFC of Female Mice

To investigate whether the receptor ERß mediates acute sEPSCs, we used the ERß-selective agonist DPN (10 nM). The overall effect of ERß activation on sEPSC amplitude was small, as it only changed by 3% and these results were with no variation (Figure 3a Base: 14.32 ± 0.71 pA; DPN: 14.71 ± 0.81 pA; U = 3, Z = −0.26, *p* = 0.79 *n* = 9). Surprisingly, within-cell analyses showed that not a single cell showed any significant change (Figure 3b). As with amplitude, the overall effect of ERß activation on sEPSc frequency was again small, with only a 3% change, and these results were with no variation (Figure 3c: base: 1.889 ± 0.45 Hz vs. 1.948 ± 0.48 Hz; U = 37, Z = 0.26, *p* = 0.791, *n* = 9). Again, within-cell analyses indicated that not a single cell experienced a significant change in sEPSC frequency (Figure 3d). As such, there was no correlated response to DPN between frequency and amplitude (Figure 3h).

### 3.4. Activation of ERß on Inhibitory Transmission in the PFC of Female Mice

Having ruled out an effect of ERß activation on glutamatergic (excitatory) transmission, we next determined whether ERß activation regulates GABAergic (inhibitory) transmission. Overall, activation of ERß reduced the sIPSC amplitude by 10% (in a DPN-treated slice as compared to a control slice; Figure 4a Base: 37.66 ± 2.527 pA; DPN: 33.92 ± 2.09 pA, *n* = 14), but these results were with no variation (Figure 4a *t* = 1.715, *df* = 13 *p* = 0.11, *n* = 14). In contrast this overall observed decrease in sIPSC in response to DPN, the within-cell analyses showed that DPN significantly increased the sIPSC amplitude in 3 of 14 cells, with increases ranging from 9% to 17%, whereas DPN significantly decreased the sIPSC amplitude in 4 of 14 cells, with decreases ranging from 18% to 40% (Figure 4b); in 7 of 14 cells, DPN had no effect on sIPSC amplitude. For sIPSC frequency responses, overall DPN increased the sIPSC frequency by 6% (Figure 4c Base: 4.054 ± 0.51 Hz, DPN: 4.313 ± 0.45 Hz, *n* = 14), but again these results were with no variation (Figure 4c U = 82, Z = −0.71, *p* = 0.47, *n* = 14). However, within-cell analyses showed that ERß activation via DPN significantly increased the sIPSC frequency in 4 of 14 cells, with increases from 14% to 102%, and significantly decreased the sIPSC frequency in 3 of 14 cells, with decreases from 9% to 26% (Figure 4d). Overall, the proportion of cells responding to DPN with an increase in sIPSC amplitude was 29% and with a decrease was 21% (Figure 4e); the proportion of cells responding to DPN with an increase in sIPSC frequency was 36% and with a decrease was 21% (Figure 4f). Similar to in other experiments, relationships between frequency and amplitude could not be seen (Figure 4h). Quantification of the kinetics of sIPSCs revealed that decay time was prolonged compared to control condition (Table 1, Base: 42.2 ± 7.2 ms, DPN: 52.5 ± 4.5 ms; * *p* < 0.05)

### 3.5. Activation of ERß on Glutamatergic Transmission in PFC of Male Mice

We next investigated whether male mice show any changes in the excitatory synaptic transmission upon ERß activation and whether this differs from the findings in females. In male mice, sEPSC amplitude measurements in response to DPN were opposite of the results found in females. In males, the overall effect of ERß activation on sEPSC amplitude was reduced by 2% (Figure 5a Base: 21.12 ± 4.01; DPN: 20.71 ± 4.22 *n* = 14), but these results were with no variation (*t* = 0.74, *df* = 13, *p* = 0.47 *n* = 14). The within-cell analyses showed that the sEPSC amplitude did not respond to DPN in 12 of 14 cells, whereas 1 cell showed a significant 9% decrease, and 1 cell showed a significant 27% increase in sEPSC amplitude (Figure 5b). Thus, of the 14 cells, 7% responded to DNP with a significant increase in sEPSC amplitude and 7% responded with a significant decrease. Regarding sEPSC frequency, overall DPN treatment increased the sEPSC frequency by 45% (Figure 5c Base: 0.97 ± 0.23, DPN 1.41 ± 0.37), but these results were with no variation (Figure 5c U = 82, Z = −0.72 *p* = 0.48). Within-cell analyses showed that ERß activation by DPN significantly increased sEPSC frequency in 6 of 14 cells (43% of cells), with increases ranging from 2% to 76%, whereas it significantly decrease sEPSC frequency in 1 of 14 cells (7% of cells), with a decrease of 88%; and 7 of 14 cells (50% of cells) showed no change in sEPSC frequency (Figure 5d,f). As the sample size was relatively small, we observed only one of 14 cells changed significantly in a subgroup of cells, thus we cannot determine whether one cell was indeed the effects on the cells, or some other mechanisms and this change needs further study. No correlation was found between changes in amplitude and frequency (Figure 5h).

### 3.6. Activation of ERß on GABAergic Transmission in PFC of Male Mice

To investigate whether activation of ERß plays a role in GABAergic (inhibitory) synaptic transmission in males, we further tested DPN on the sIPSC amplitude and frequency in males. As shown in Figure 6a, the overall effect of DPN increased the sIPSC amplitude by 4% (Figure 6a Base: 25.42 ± 1.74; DPN: 26.33 ± 1.60 *n* = 14), but these results were with no variation (Figure 6a U = 78, Z = 0.90, *p* = 0.37). Yet, the within-cells analyses showed that 4 of 14 cells (29% of cells) responded with significant sIPSC amplitude increases, ranging from 7% to 20%, whereas 3 of 14 cells (21% of cells) responded with significant sIPSC amplitude decreases, ranging from 6% to 12%; 7 of 14 cells (50% of cells) responded to DPN with no effect on sIPSC amplitude (Figure 6b). Regarding sIPSC frequencies, we found that overall, ERß activation via DPN increased sIPSC frequency by 8% (Figure 6c Base: 2.40 ± 0.39, DPN: 2.60 ± 0.47; U = 95, Z = −0.11, *p* = 0.91, *n* = 14). Considering the within-cell analyses, 5 of 14 cells (36% of cells) responded with significant increases in sIPSC frequency, ranging from 16% to 240%, whereas 4 of 14 cells (28% of cells) responded with significant decreases in sIPSC frequency, ranging from 19% to 89% (Figure 6d,f). Meanwhile, no correlations were found between sIPSC frequency and amplitude changes in response to DPN (Figure 6h). Quantification of the kinetics of sIPSCs revealed that decay time was prolonged compared to control condition (Table 2, Base: 25.0 ± 1.8 ms, DPN: 31.9 ± 2.8 ms; * *p* < 0.05).

### 3.7. Estradiol Enhances Excitatory Synaptic Transmission in a Subgroup of Cells in the Hippocampus of Female Mice

In light of the various effects of estradiol on synaptic transmission in the PFC, we next determined whether E2 affects excitatory synaptic transmission in pyramidal neurons of the CA1 region of females and males. As shown in Figure 7a, estradiol (E2) caused a significant increase in sEPSC amplitude in comparison to the control condition (Figure 7a; Base: 14.53 ± 0.50 pA Vs. 13.46 ± 0.50 pA, *t* = 2.313, *df* = 9, *p* = 0.046, *n* = 10). Within-cell analyses indicated that the overall effect of E2 was small, as only 3 of 10 cells (30%) showed significant increases in sEPSC amplitude, with increases ranging from 11% to 32% (Figure 7b). Conversely, 7 of 10 cells (70%) showed no signifiant response to E2 in sEPSC amplitude (Figure 7e). For sEPSC frequency, the overall effect of E2 did change sEPSC frequency, but these results were with no variation (Figure 7c U = 39, Z = −0.793, *p* = 0.42). The within-cell analyses showed that E2 significantly increased the sEPSC frequency in 4 of 10 cells (40%), with increases ranging from 41% to 528% (Figure 7d,f). Regarding normalized effects, the plot in Figure 7h shows that effects typically occurred in different cell subsets in female mice. Interestingly, only 1 of 10 cells responded to E2 with changes in both sEPSC frequency and amplitude (Figure 7h). These experiments suggest that in female mice, E2 increases sEPSC amplitude and frequency, but often in different subsets of cells.

### 3.8. Estradiol Enhances Excitatory Synaptic Transmission in a Subgroup of Cells in the Hippocampus of Male Mice

In males, the overall effect of E2 was altered the sEPSC amplitude of hippocampal cells by 1.6%, but these results were with no variation (Figure 8a Base: 15.41 ± 0.84 pA, vs. 15.66 ± 1.04 pA; *t* = 0.586, *df* = 7, *p* = 0.576, *n* = 8). Within-cell analyses showed that E2 significantly increased the sEPSC amplitude in 3 of 8 cells (38%), with increases ranging from 7% to 11% (Figure 8b,e). For sEPSC frequency, overall E2 did show a change, but these results were with no variation (Figure 8c, Base: 3.18 ± 3.50; E2 3.80± 3.72; U = 28, Z = −0.367, *p*= 0.713). Within-cell analyses showed that E2 increased the sEPSC frequency in 3 of 8 male hippocampal cells (38%), with increases ranging from 17% to 188% (Figure 8d,f). Interestingly, although E2 increased both the sEPSC frequency and amplitude in both sexes, the within-cell analyses showed that these effects occurred in different subsets of cells. As with the females, again there was no significant correlation between the relative frequency and relative amplitude changes in male hippocampal cells (Figure 8h).

## 4. Discussion

In the current study, we investigated the effect of estradiol on synaptic transmission in the PFC and hippocampus of female and male mice. The above data demonstrate that (1) E2 acts as a neuromodulator of spontaneous signal transduction in the PFC of both sexes and has an effect on synaptic transmission. (2) The response to E2 in PFC can be divided into three different types: no significant change; significant increase; significant decrease. (3) In both sexes, E2 affects sEPSc frequencies and amplitude through a mix of increases and decreases. (4) In the female PFC, activating ERß via DPN did not influence excitatory transmission. (5) However, activating ERß via DPN in female PFC had various effects on inhibitory transmission, namely on sIPSC amplitude and frequency. Kinetics analyses showed that decay time is strongly prolonged in both male and female mice. (6) E2 increases sEPSCs amplitude significantly in female hippocampus, but not in male.

### 4.1. The Response to E2 in PFC Causes an Different Reaction in Synaptic Transmission in Both Sex

In both females and males, E2 led to significant changes in sEPSC amplitude and frequency in the PFC. In particular, we observed that E2 affected synaptic transmission either by increasing or decreasing the frequency and/or amplitude. Previous studies have shown that E2 can affect the synaptic activity in various brain regions [3,18,25,26]. However, in our experiments, the within-cell analyses showed that responses to E2 take place within minutes but not all cells respond with significant changes in amplitude and/or frequency. Further, some cells show no response to E2. This leads to the question of how a single active substance can cause opposite reactions in different cells under the same conditions. One possible reason might be of localization specificity. For example, the mode of action of erythropoietin (EPO) on neurotransmission in the PFC area differs depending on the hemisphere under investigation [27].

While studies on the hippocampal region have shown that the different effects of E2 are mostly mediated by different ER subtypes [3,25,28], for the PFC E2 simultaneously affects different ER subtypes and further leads to amplitude and/or frequency responses in opposite directions. As shown in Figure 4, sIPSCs show opposing responses in the PFC of female mice.

Again, activating ERß via DPN proportionally increased and decreased the sIPSC amplitudes and frequencies (Figure 4 and Figure 6). To determine whether the distribution of increases and decreases in amplitude and frequency are indeed unequal in females versus males, one would need a larger sample set.

We observe the response to DPN caused an opposite reaction on sIPSC frequency and amplitude within cells analyses in PFC (Figure 4). It has been found that Eα present especially in those interneurons that express cholecystokinin and colocalization with neuropeptide Y, but not parvalbumin, whereby estrogen associated with cluster of vesicles in synaptic boutons and mobilizes these vesicles clusters toward synapses and increase release neurotransmitter in hippocampus and leads to increase the sIPSC amplitude or frequencies [29]. Moreover, estradiol has been found to regulate the levels of mRNA for both forms of GAD (GAD65 and GAD67) in various regions [30]. In contrast E2 potentiation of GABAergic transmission, E2 treatment reduces the number of synaptic GABAa Rs and Gephyrin, via a postsynaptic mechanism that relies on disrupting the postsynaptic scaffold and in the end attenuate inhibitory synaptic transmission [31]. It further leads to reduced sIPSC amplitude or frequency.

Moreover, the measurement of kinetic showed that there are no variation in the half-width, but we observed an averaged decay prolongation (Figure 4 and Figure 6), suggesting an involving the slow tau decay. Considering no variation in half-width, the response to DPN in IPSC may be explained by the changes in GABA_A_ receptor in the subunit composition [32].

### 4.2. ERß Activation Especially Modulates Glutamatergic Transmission in the Hippocampus

In contrast to E2 responses in the PFC, our data from the hippocampus showed that the E2 acutely potentiates glutamatergic transmission in sEPSC amplitudes and frequencies in both sexes. These observations are consistent with studies that found estradiol acutely potentiates glutamatergic transmission in the hippocampus through a presynaptic mechanism by increasing the probability of glutamate release in female rats [3]. Meanwhile, those effects are mediated by ERß acting as a monomer, whereas ERα is not required. These authors observed that only some CA1 cells responded to ERß, suggesting that ERß activation is cell specific [3]. Interestingly our hippocampus data showed that there was no significant decrease sEPSC amplitude and frequency in hippocampus, which differs from our results in the PFC, suggesting that responses to ERß activation are likely region specific and various signaling pathway.

### 4.3. E2 and ERß Activation Have Both Potentiating and Suppressive Effects on Excitatory Transmission in the PFC

In males and females, our present results showed that E2 has different effects on the sEPSC amplitude and frequency. Remarkably, within-cell analyses showed that E2-responsive and non-responsive cells were found in both sexes. Despite the specific investigation of excitatory transmission, significant increases and decreases in both amplitude and frequency were still found in males, especially for the sEPSC amplitude, which was significantly increased in 12 of 24 responsive male cells in the PFC, whereas for females, 3 of 11 cells in the PFC showed significant sEPSC amplitude changes. In terms of frequency, we found that increases and decreases in sEPSC frequency occurred in almost 50% of cells. However, in females the PFC data showed that only 3 of 14 cells had significant increases in sEPSC frequency, and the majority of cells were not responsive to E2.

Activation of ERß through DPN, a selective ERß agonist, affected synaptic transmission in various way for males and females. Nonetheless, these data suggest that E2 tends to modulate excitatory synaptic transmission in various ways between males and females. It is likely that E2 potentiates excitatory synaptic transmission through a presynaptic mechanism and by increasing the probability of glutamate release at a synapse [3]. In neurons, E2 activates L-type Ca^2+^ channels and acutely increase Ca^2+^ influx and further activates protein kinase signaling pathways, including Src, Erk172, CamKII and the cAMP cascade [33,34], whereby these intracellular signaling cascades modulate synaptic transmission, synaptic plasticity, channels and transcription factors [35]. Consistently, we demonstrate that E2 altered the sEPSCs frequency via a presynaptic mechanism.

E2 enhances sEPSC amplitude is probably thought to result from enhancing NMDA receptors and increasing membrane levels of AMPA receptors by the mitogen-activated protein kinase (MAPK)-mediated phosphorylation and cAMP-response element-binding protein (CREB) [36,37]. It has been shown that E2 binds to membrane-associated estrogen receptors, activating both group I and II mGluR signaling in a glutamate-independent manner, which results in bidirectional signaling pathway [38,39]. Group I consist of mGluR1 and mGluR5, which are Gq linked, and its stimulation through E2 leads to MAPK-dependent CREB phosphorylation via activation of phospholipase (PLC), protein kinase C (PKC) and inositol triphosphate (IP3) signaling, furthering CREB phosphorylation [39,40].

Conversely Estradiol also activates group II mGluRs, namely mGluR2 and mGluR3, via ERα or ß, which are linked to the Gi/o singling pathway, leading to diminished L-type calcium channel-dependent CREB phosphorylation and decreased cAMP concentrations and a reduction in PKA activity.

Early studies have found that the action of estradiol in the brain is sex specific, as are the actions of estradiol on mGluR signaling [41,42]. Similarly, in current studies we observed that E2 responses caused opposite reactions in sEPSC amplitudes and frequencies in the PFC of female and male mice. While ER/mGluR mediates several sex-specific estradiol actions on hippocampal neuron function, mGluR action itself is not sex-specific. However, Meitzen and colleagues found that caveolin showed different sex-specific expression in the adult hippocampus [43], such that the genes necessary for the ER membrane complex, including those that encode ERα and ERß, caveolin 1 and 3, are likely differently expressed in males and females [44]. Furthermore, the expression of caveolin 1 and DHHC-7 are reduced in the adult hippocampus in females as compared to in males [43].

Altogether, estradiol most likely elicits synaptic neurotransmission changes not only in a distinct signaling pathway but also in a sex specific way.

## Figures and Tables

**Figure 1 ijms-22-01485-f001:**
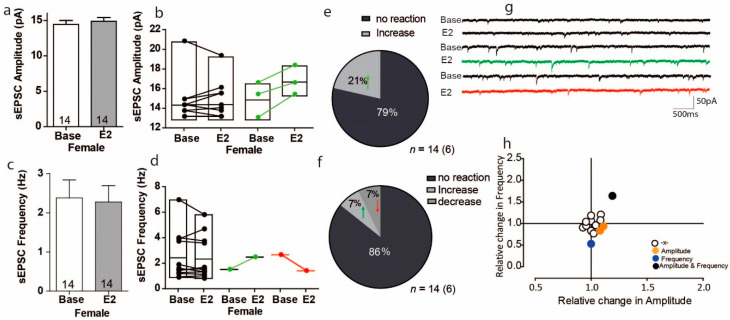
Estradiol causes a reaction in a subpopulation of cells in the PFC of female mice. (**a**) Average amplitude of sEPSC recordings in mPFC during baseline and after E2 treatment. (**b**) Plots showing individual sEPSC amplitude responses to E2 during the baseline and after E2 treatment in the same cell. 3 of 14 cells in the within-cell analyses showed significant increases after E2 treatment. Unless noted, colored symbols represent the subset of cells in which within-cell statistical analyses showed significant effect of E2. Green connected symbol represents cells that had a significant increase in response to E2. (**c**) The overall response to E2 on sEPSC frequency in mPFC during baseline and after E2 treatment. (**d**) Plots showing sEPSC frequency during baseline and after E2 treatment within-cell analyses. Unless noted, red connected symbol represents cells that had a statistically significant decrease in response to E2. (**e**,**f**) The proportion of cells with a significant response to E2 treatment (**g**). Sample traces, frequencies and amplitudes of sEPSCs before and after E2 treatment (**h**). Plots of the normalized change in sEPSC frequency versus amplitude for each cell; E2 treatment hardly changed the sEPSC frequency and amplitude in the same cell.

**Figure 2 ijms-22-01485-f002:**
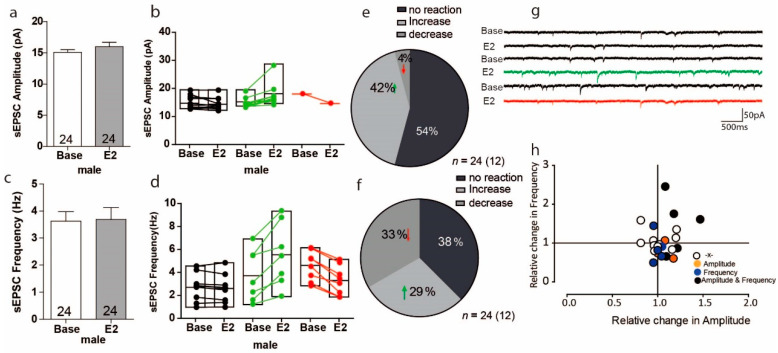
Estradiol causes an opposing reaction in a subset of cells in the PFC of male versus female mice. (**a**) No significant change was found in the average amplitude of sEPSC recordings in the mPFC during baseline and after E2 treatment. (**b**) Plots showing individual sEPSC amplitude responses to E2 during the baseline and after E2 treatment in the same cell. (**c**) The overall response to E2 regarding sEPSC frequency in the mPFC during baseline and after E2 treatment. (**d**) Plots showing sEPSC frequency during baseline and after E2 treatment within-cell analyses. (**e**,**f**) The proportion of cells with a significant response to E2 treatment. (**g**) Sample traces, frequencies and amplitudes of sEPSCs before and after E2 treatment. (**h**) Plot of the normalized change in sEPSC frequency versus amplitude for each cell; E2 treatment changed sEPSC frequency and amplitude in various ways in different cells.

**Figure 3 ijms-22-01485-f003:**
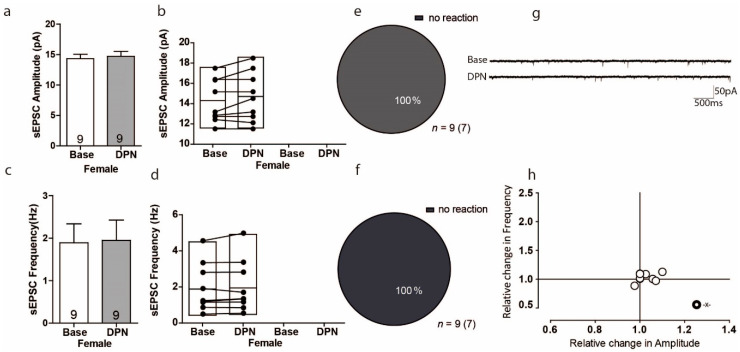
Activation of ERß had no influence on excitatory transmission in the PFC of female mice. (**a**,**c**) No significant changes were found in the average sEPSC amplitude and sEPSC frequency in mPFC during baseline and after E2 treatment. (**b**,**d**) Plots showing individual sEPSC amplitude responses to E2 within-cell analyses during the baseline and after E2 treatment in the same cell. (**e**,**f**) The proportion of cells responding to E2 treatment. (**g**) Sample traces, frequencies and amplitudes of sEPSCs before and after E2 treatment. (**h**) Plots showing the normalized change in sEPSC frequency versus amplitude for each cell.

**Figure 4 ijms-22-01485-f004:**
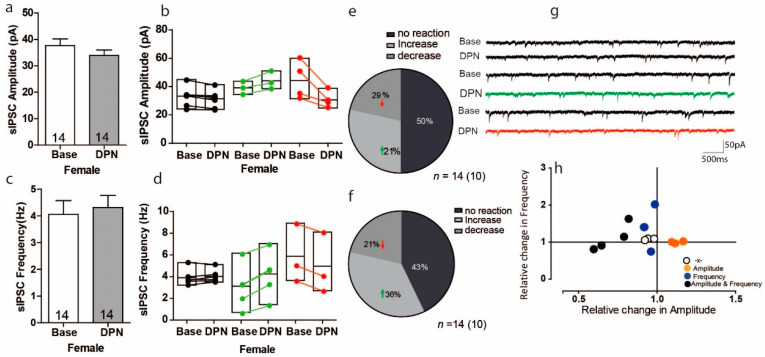
Activation of ERß on inhibitory transmission in the PFC of female mice. (**a**,**c**) No significant changes were found in the average sIPSC amplitude and sEPSC frequency recordings in the mPFC during baseline and after E2 treatment. (**b**,**d**) Within-cell analyses show that activation of ERß led to opposing reactions on sIPSC amplitude and frequency in a subset of cells. (**e**,**f**) The proportion of cells with a significant response to DPN treatment. (**g**) Sample traces, frequencies and amplitudes of sEPSCs before and after DPN treatment. (**h**) No correlations were found between changes in sIPSC frequency and sIPSC amplitude.

**Figure 5 ijms-22-01485-f005:**
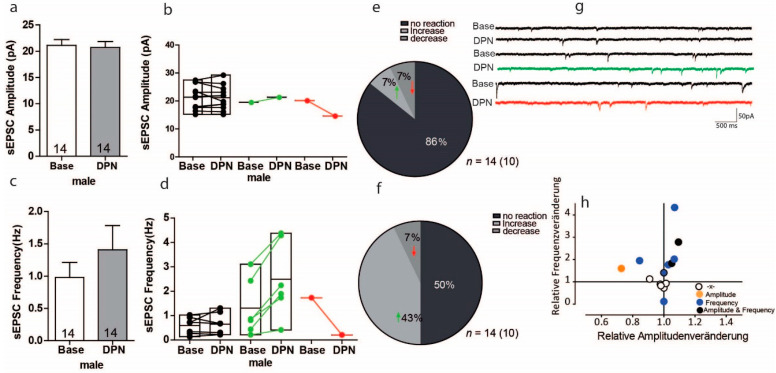
Activation of ERß on glutamatergic transmission in the PFC of male mice. (**a**,**c**) No significant changes in the average sIPSC amplitude and sEPSC frequency recordings were found during baseline and after DPN treatment. (**b**,**e**) Within-cell analyses show the majority of 14 cells had no significant response to DPN in sEPSC amplitude. (**d**,**f**) 6 of 14 cells showed a significant response to DPN treatment in sEPSC frequency. (**g**) Sample traces, frequencies and amplitudes of sEPSCs before and after DPN treatment. (**h**) No correlations were found between changes in sIPSC frequency and sIPSC amplitude.

**Figure 6 ijms-22-01485-f006:**
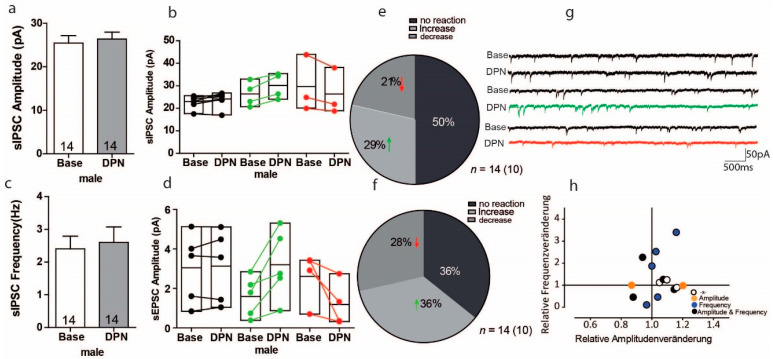
Activation of ERß on inhibitory transmission in the PFC of male mice. (**a**,**c**) No significant changes in the average sIPSC amplitude and sEPSC frequency recordings were found in the mPFC during baseline and after DPN treatment. (**b**,**d**) Within-cell analyses showed that activation of ERß led to opposing reactions in the sIPSC amplitude and frequency in a subset of cells. (**e**,**f**) The proportion of cells with a significant response to DPN treatment. (**g**) Sample traces, frequencies and amplitudes of sEPSCs before and after DPN treatment. (**h**) No correlations were found between changes in sIPSC frequency and sIPSC amplitude.

**Figure 7 ijms-22-01485-f007:**
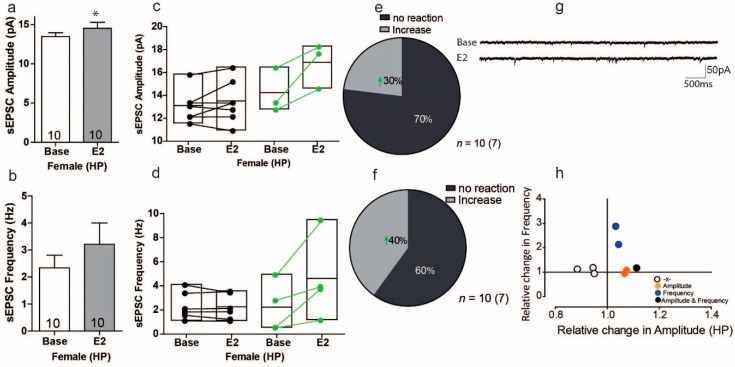
Estradiol enhances sEPSC amplitude and frequency in a subgroup of cells in the hippocampus of female mice. (**a**,**c**) Significant changes were found in the average sEPSC amplitudes but not in the sEPSC frequency recordings during baseline and after E2 treatment. (**b**,**d**) Within-cell analyses showed that some cells significantly responded to E2 in sEPSC amplitude and sEPSc frequency. (**e**,**f**) The proportion of cells that responded to E2 treatment in sEPSC amplitude and sEPSC frequency. (**g**) Sample traces, frequencies and amplitudes of sEPSCs before and after DPN treatment. (**h**) No correlations between changes in sIPSC frequency and sIPSC amplitude were found; * HP, Hippocampus.

**Figure 8 ijms-22-01485-f008:**
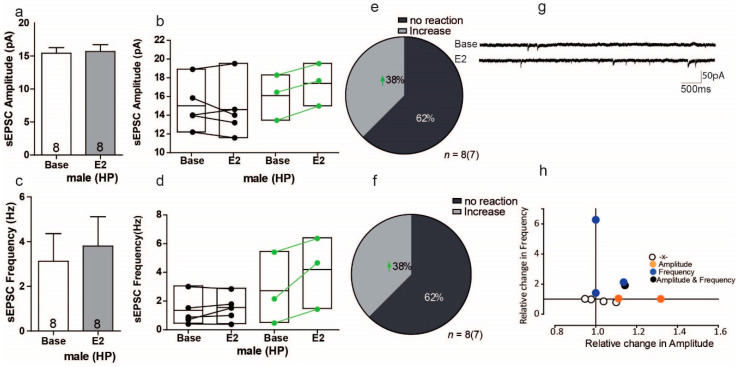
Estradiol enhances sEPSC frequency and amplitude in a subgroup of cells in the hippocampus of male mice. (**a**,**c**) No significant changes were found in the average sEPSC amplitude and sEPSC frequency recordings during baseline and after E2 treatment. (**b**,**d**) Within-cell analyses showed that some cells significantly responded to E2 in sEPSC amplitude and sEPSc frequency. (**e**,**f**) The proportion of cells that responded to E2 treatment in sEPSC amplitude and sEPSC frequency. (**g**) Sample traces, frequencies and amplitudes of sEPSCs before and after DPN treatment. (**h**) No correlations were found between changes in sIPSC frequency and sIPSC amplitude; HP, Hippocampus.

**Table 1 ijms-22-01485-t001:** Kinetics of spontan GABAergic currents in Base and after application of DPN in females.

aIPSC	Deca Time (ms)	Half-Width (ms)	Area (pA × ms)
Base	42.2 ± 7.2	13.3 ± 0.5	968 ± 74.8
DPN	52.5 ± 4.5 *	12.6 ± 0.6	1102 ± 104.7 *

**Table 2 ijms-22-01485-t002:** Kinetics of spontan GABAergic currents in Base and after application of DPN in males.

sIPSC	Decay Time (ms)	Half-Width (ms)	Area (pA × ms)
Base	25.0 ± 1.8	8.9 ± 0.4	724.8 ± 52.5
DPN	31.9 ± 2.8 *	9.3 ± 0.5	908.7 ± 85.0

## Data Availability

The raw data presented in this study are available on request from the corresponding autor.

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
