# Peer review of "Estrogen Receptors Alpha and Beta Mediate Synaptic Transmission in the PFC and Hippocampus of Mice"

_ijms, 2021, doi:10.3390/ijms22031485_

Round 1

Reviewer 1 Report

Zhang and collaborators investigate the involvement of estradiol in the modulation of excitatory and inhibitory synaptic transmission in two brain structures: PFC and hippocampus. They observed a heterogeneity in the synaptic transmission after estradiol application and a gender-based difference on spontaneous synaptic transmission. E2 action is partially mediated through ER beta activation.

The results currently presented are interesting but seems premature and need to be strengthened. Moreover part of it is over-interpreted.

First, authors claim to observe a non-significant variation but call it a decrease or an increase. However a non-significant difference, is not a difference, especially when p value is far from significance threshold, it cannot even been called a trend.

Then based on individual cell analysis, they claim that some cells have significant individual variations without showing any numerical value or explaining it in M & M. To strengthen this point many complementary experiments or analysis are needed:

- Classically to ascertain such individual variation, authors doesn’t used a statistically analysis with p>0.05 but use a more robust criterion such as a variation greater than 20-25% of control.

- To confirm such individual variation authors should also prove that the patched population is homogen. Indeed in M & M, authors wrote that recordings are made in layer 2/3 without stating the type of cell patched. Moreover homogeneity could be demonstrated through small dispersion of passive membrane properties (membrane resistance and capacitance, at least) parameters values.

- A parameter of PSC is not investigated: kinetics (especially tau decay). Indeed E2 could act as a neurosteroid and on GABAa synaptic transmission neurosteroid are known to strongly prolong tau decay. This action as an neurosteroid is reinforced by the delay needed to observe change.

- In the present work variations are highly variable and not robust between frequency and amplitude. Classically a way to pool together variations in amplitude, frequency and kinetics is to measure the area under the curve (AUC) and to compare results as cumulative representation with Kolmogorov test.

These results are obtained for spontaneous synaptic transmission but should be replicated with miniature to fine tune the conclusions.

In addition in some cases authors observed variations in only ONE cell out of 14 (for example) and strongly interpret it. I would advise to increase the sample size. Moreover in figure 1d (for example) bar graph shows that a non-changing cell seems to have the same reduction in frequency than a decreasing one! Likewise in figure 3 variability in DPN condition seems equivalent to under E2 where this variation have been classified as an increase.

The title is not in agreement with the pharmacological experiments performed, only ER beta agonist was used. Moreover in pharmacological experiment effect of an agonist is supported by the co-application of agonist and antagonist.

Author Response

We appreciated the reviewer’s Comments. The followings are point by point response:

Point1: First, authors claim to observe a non-significant variation but call it a decrease or an increase. However a non-significant difference, is not a difference, especially when p value is far from significance threshold, it cannot even been called a trend.

Response: Thank you for pointing out these mistakes, we revised them in results part (line 160, 168, 205, 209, 243, 250, 289, 300, 301, 304,)

Point2: Then based on individual cell analysis, they claim that some cells have significant individual variations without showing any numerical value or explaining it in M & M. To strengthen this point many complementary experiments or analysis are needed:

Response: we agree with the reviewers Comment, however a similar methodological analysis was published by Oberlander and his colleague (JG Oberlander & CS Woolley 2016).

Oberland JG, & Woolley CS. (2016). 17β-Estradiol Acutely Potentiates Glutamatergic Synaptic Transmission in the Hippocampus through Distinct Mechanisms in Males and Females. 36(9): 2677–2690.

Point 3- Classically to ascertain such individual variation, authors doesn’t used a statistically analysis with p>0.05 but use a more robust criterion such as a variation greater than 20-25% of control.

Response:  many thanks for your comments. We like to clarify the way we analyzed the data.  For individual cell analysis we first test if the events of recordings if they are normally distributed, and if they are, we performed unpaired Two-tailed t test with welch’s correction, and if they are not, we used the Two-tailed Mann-Whitney test. Besides statistical analysis, we additionally described the individual cells’ response to Estradiol or DPN in terms of percentages to response to the drug to give the reader more intuitive results.

Point4- To confirm such individual variation authors should also prove that the patched population is homogen. Indeed in M & M, authors wrote that recordings are made in layer 2/3 without stating the type of cell patched. Moreover homogeneity could be demonstrated through small dispersion of passive membrane properties (membrane resistance and capacitance, at least) parameters values.

Response: Thank you very much for your advices. Under the microscope we can clearly see the morphology of the cells and avoid to patch interneurons in PFC. In our lab we have been working in interneurons in PFC for a long time. We did a lot relevant immunohistochemical experiment for interneurons in PFC, including Palvalbumine (PV), Somatostatin,  Calbindine and calretinie etc . Part of results plus Patch clamp recordings we have already published (R Safari et al 2019). Such experiences could better help us avoid patching interneurons in PFC and in hippocampus.  Besides these, during the Patching process we also monitor access resistance of neurons and we noted parameters of the membrane capacitance and resting potential. If the resting potential exceed -60mV it will be excluded. In the end we limit the Pyramidal neurons II/III layers as much as possible to ensure homogeneity of the patched Pyramidal neurons.We included more relevant details information in the materials and methods section on page 5 line 101-109

Point5- A parameter of PSC is not investigated: kinetics (especially tau decay). Indeed E2 could act as a neurosteroid and on GABAa synaptic transmission neurosteroid are known to strongly prolong tau decay. This action as an neurosteroid is reinforced by the delay needed to observe change.

Response: we performed new kinetic analysis and added the new table 4.1 in the Figure4 and table 6.1 in the figure6 and kinetics analysis on page 6 127-138

Point6- In the present work variations are highly variable and not robust between frequency and amplitude. Classically a way to pool together variations in amplitude, frequency and kinetics is to measure the area under the curve (AUC) and to compare results as cumulative representation with Kolmogorov test.

Response: we observed the effects of estrogen receptor mediated synaptic transmission was most of in a subgroup of the cells, and the some response in an opposite way , so we chose to show the effect of the drug in the individual cells in the end. However, we appreciate your suggestions and we will follow them next time if we encounter a similar situation as well.

Point7These results are obtained for spontaneous synaptic transmission but should be replicated with miniature to fine tune the conclusions.

Response: We totally agree with you. Action potential independent miniature synaptic transmission provides further more information about presynaptic and postsynaptic mechanisms. In this article, we mainly focus on the effects of Estradiol and DPN on sEPSC and sIPSC of cells of PFC, hippocampus and difference between genders as well. For miniature synaptic transmission we would like to put the follow up studies.

Point8a In addition in some cases authors observed variations in only ONE cell out of 14 (for example) and strongly interpret it. I would advise to increase the sample size.

Response: we thank reviewer for this valuable comments. We agree with reviewers comments that we need a large sample size in order to continue and confirm this changing phenomenon. Given the current state of the pandemic, we have here very strict rules in the hospital and in the animal facility, which will make it more difficult for us to do some additional experiments. We hope you can understand it.  In addition, according your comments, I revised and clarify the issue in the text in the page 8 Line 172-174 and page 10 line 255-258.

Point8b Moreover in figure 1d (for example) bar graph shows that a non-changing cell seems to have the same reduction in frequency than a decreasing one! Likewise in figure 3 variability in DPN condition seems equivalent to under E2 where this variation have been classified as an increase.

Response: In light of your comments, we rechecked the raw data and statistical analysis we used for the both graphs and results remain unchanged.

Point9 The title is not in agreement with the pharmacological experiments performed, only ER beta agonist was used. Moreover in pharmacological experiment effect of an agonist is supported by the co-application of agonist and antagonist.

Response: our drug E2 (ß-estradiol from Tocris) information will be included in the Methods. The drug ß-estradiol (Tocris) is unspecific can activate both alpha and beta estrogen receptor and DPN is a selective agonist of ERß receptors. In addition, we agree with your comments and suggestions by the co-application of agonist and antagonist. We will keep this in mind for future experiments

Reviewer 2 Report

The topic of the manuscript is interesting but should be improved by information how drug application was realized?. English need improvement before publication.

Section "Reference" need by carefully check. Main errors: duplicate of citations and missing of first letter.

Hu W [et al.] tress impairs GABAergic network function in the hippocampus by 629 activating nongenomic glucocorticoid

Weller M und Waltereit R signaling from cAMP/PKA to MAPK and synaptic 715 plasticity [Journal]. - [s.l.] : Mol Neurobiol, 2003. - Bde. 27: 99-106.. 716

Weller M und Waltereit R Signaling From cAMP/PKA to MAPK and Synaptic 717 Plasticity [Journal]. - [s.l.] : Mol Neurobiol, 2003. - Bde. 27(1):99-106.

Wu TW [et al.] 1β-estradiol induced Ca2+ influx via L-type calcium channels

Arnold AP und Breedlove SM rganizational and activational effects of sex 602 steroids on brain and behavior: a reanalysis. [Journal

Fuster JM The prefrontal cortex - An update: Time is of the essence. [Journal]. - 616 [s.l.] : Neuron., 2001. - Bde. 30:319-333. 617

Fuster MJ The prefrontal cortex-an update: Time is of the essence [Journal]. - 618 [s.l.] : Neuron, 2001. - Bde. 30,319-333.

Smejkalova T und Wooley CS Estradiol acutely potentiates hippocampal 688 excitatory synaptic transmission through a presynaptic mechanism [Journal]. - 689 [s.l.] : J Neurosci, 2010. - Bde. 30(48):16137–48. 690

Smejkalova T und Woolley CS Estradiol acutely potentiates hippocampal 691 excitatory synaptic transmission through a presynaptic mechanism. [Journal]. - 692 [s.l.] : J Neurosci Society for Neuroscience, 2010. - Bde. 30: 16137–48.

Tabatadze N [et al.] Sex Differences in Molecular Signaling at Inhibitory Synapses 694 in the Hippocampus [Journal]. - [s.l.] : J Neurosci., 2015. - Bde. 35(32): 11252–695 11265.. 696

Tabatadze N [et al.] Sex Differences in Molecular Signaling at Inhibitory Synapses 697 in the Hippocampus. [Journal]. - [s.l.] : J Neurosci Society for Neuroscience, 2015. - 698 Bde. 35: 11252–65.

Weller M und Waltereit R signaling from cAMP/PKA to MAPK and synaptic 715 plasticity [Journal]. - [s.l.] : Mol Neurobiol, 2003. - Bde. 27: 99-106.. 716

Weller M und Waltereit R Signaling From cAMP/PKA to MAPK and Synaptic 717 Plasticity [Journal]. - [s.l.] : Mol Neurobiol, 2003. - Bde. 27(1):99-106.

Author Response

Point1The topic of the manuscript is interesting but should be improved by information how drug application was realized?. English need improvement before publication.

Response: we are appreciated for your time and comments. As suggested the drug information is now included on page 5 line 124-126. We asked for a proofreading by our science writing support service (SWS,University Muenster) before submission.

Point2Section "Reference" need by carefully check. Main errors: duplicate of citations and missing of first letter.

Hu W [et al.] tress impairs GABAergic network function in the hippocampus by 629 activating nongenomic glucocorticoid

Weller M und Waltereit R signaling from cAMP/PKA to MAPK and synaptic 715 plasticity [Journal]. - [s.l.] : Mol Neurobiol, 2003. - Bde. 27: 99-106.. 716

Weller M und Waltereit R Signaling From cAMP/PKA to MAPK and Synaptic 717 Plasticity [Journal]. - [s.l.] : Mol Neurobiol, 2003. - Bde. 27(1):99-106.

Wu TW [et al.] 1β-estradiol induced Ca2+ influx via L-type calcium channels

Arnold AP und Breedlove SM rganizational and activational effects of sex 602 steroids on brain and behavior: a reanalysis. [Journal

Fuster JM The prefrontal cortex - An update: Time is of the essence. [Journal]. - 616 [s.l.] : Neuron., 2001. - Bde. 30:319-333. 617

Fuster MJ The prefrontal cortex-an update: Time is of the essence [Journal]. - 618 [s.l.] : Neuron, 2001. - Bde. 30,319-333.

Smejkalova T und Wooley CS Estradiol acutely potentiates hippocampal 688 excitatory synaptic transmission through a presynaptic mechanism [Journal]. - 689 [s.l.] : J Neurosci, 2010. - Bde. 30(48):16137–48. 690

Smejkalova T und Woolley CS Estradiol acutely potentiates hippocampal 691 excitatory synaptic transmission through a presynaptic mechanism. [Journal]. - 692 [s.l.] : J Neurosci Society for Neuroscience, 2010. - Bde. 30: 16137–48.

Tabatadze N [et al.] Sex Differences in Molecular Signaling at Inhibitory Synapses 694 in the Hippocampus [Journal]. - [s.l.] : J Neurosci., 2015. - Bde. 35(32): 11252–695 11265.. 696

Tabatadze N [et al.] Sex Differences in Molecular Signaling at Inhibitory Synapses 697 in the Hippocampus. [Journal]. - [s.l.] : J Neurosci Society for Neuroscience, 2015. - 698 Bde. 35: 11252–65.

Weller M und Waltereit R signaling from cAMP/PKA to MAPK and synaptic 715 plasticity [Journal]. - [s.l.] : Mol Neurobiol, 2003. - Bde. 27: 99-106.. 716

Weller M und Waltereit R Signaling From cAMP/PKA to MAPK and Synaptic 717 Plasticity [Journal]. - [s.l.] : Mol Neurobiol, 2003. - Bde. 27(1):99-106.

Response: revised as suggested.

Round 2

Reviewer 1 Report

Authors convincingly answered to most of my comments, however the corrections they made in the text in response to the first one (non-significant variations are not variations) are insufficient. The absence of overall variation does not weaken the results because it’s clearly explained by averaging of opposite variations. Therefore all reference to any non-significant change has to be replaced by “no variation” especially when p value is ranging between 0.5 and 0.7 that even excludes any trend.

More importantly, the measurement of kinetic brings a new result that authors does not exploit or discuss. Indeed in both type of IPSC, prolonged kinetic is the only overall reliable variation. Moreover as half width does not vary, it points to an explanation involving the slow tau decay. But authors does not discuss this point even if DPN experiments clearly exclude a direct E2 effect on GABAaR.

Author Response

Thank you very much for your comments and suggestions.  I  have marked the corrected parts in yellow in the manuscript.

Point1 Authors convincingly answered to most of my comments, however the corrections they made in the text in response to the first one (non-significant variations are not variations) are insufficient. The absence of overall variation does not weaken the results because it’s clearly explained by averaging of opposite variations. Therefore all reference to any non-significant change has to be replaced by “no variation” especially when p value is ranging between 0.5 and 0.7 that even excludes any trend.

Response: revised as suggested (Page 3-10 labeled in yellow in the results section)

More importantly, the measurement of kinetic brings a new result that authors does not exploit or discuss. Indeed in both type of IPSC, prolonged kinetic is the only overall reliable variation. Moreover as half width does not vary, it points to an explanation involving the slow tau decay. But authors does not discuss this point even if DPN experiments clearly exclude a direct E2 effect on GABAaR.

Response:  Thank you very much for your suggestions. We now added discussing information about the prolongation of decay time responded to DPN in the manuscript. It is true that we do not know whether the effects of E2 on the decay time of GABAergic receptors and is consistent with DPN response, and this point needs to further investigation.
